# Fusion Analysis of Optical Satellite Images and Digital Elevation Model for Quantifying Volume in Debris Flow Disaster

**Hiroyuki Miura** 

Department of Architecture, Hiroshima University, Higashi-Hiroshima 739-8527, Japan;
hmiura@hiroshima-u.ac.jp

**Abstract:** Rapid identification of affected areas and volumes in a large-scale debris flow disaster is important for early-stage recovery and debris management planning. This study introduces a methodology for fusion analysis of optical satellite images and digital elevation model (DEM) for simplified quantification of volumes in a debris flow event. The LiDAR data, the pre- and post-event Sentinel-2 images and the pre-event DEM in Hiroshima, Japan affected by the debris flow disaster on July 2018 are analyzed in this study. Erosion depth by the debris flows is empirically modeled from the pre- and post-event LiDAR-derived DEMs. Erosion areas are detected from the change detection of the satellite images and the DEM-based debris flow propagation analysis by providing predefined sources. The volumes and their pattern are estimated from the detected erosion areas by multiplying the empirical erosion depth. The result of the volume estimations show good agreement with the LiDAR-derived volumes.

**Keywords:** debris flow; volume; erosion depth; change detection; debris flow propagation analysis

## 1. Introduction

Landslides triggered by heavy rainfalls and earthquakes have been one of the frequently occurring natural disasters in all over the world. A landslide is defined as a movement of a mass of rock, debris or earth down a slope, and involves various slope movements such as rotational/translational landslides, rockfalls, topples, debris flows and others [1,2]. Debris flow is a form of rapid movement of loose soil, rocks and driftwoods flowed to downstream along valleys due to slope failures. Debris flows have produced severe damage to built-up areas in downstream since enormous debris largely spread to buildings, roads and other infrastructures.

Chugoku district, located in the western part of Honshu Island, Japan including Hiroshima prefecture is widely covered with weathered granites, and has been experienced by many debris flow disasters. On 29 June 1999, rainfall-induced debris flows brought damage to the residential areas with 31 fatalities in the western part of Hiroshima city [3]. On 20 August 2014, heavy rainfall-induced debris flows hit the residential areas and produced severe structural damage including 76 fatalities in the northern part of Hiroshima city [4,5]. After these debris flow disasters in Japan, the national and local governments have undertaken the disaster mitigation measures such as new construction of debris control dams, re-identification of hazardous zones and their public disclosure to citizens. At the beginning of July 2018, an extremely heavy rainfall occurred over wide areas of Japan. Despite the efforts by the governments, extensive damage was produced by huge number of flooding and landslides, including debris flows triggered by the heavy rainfalls, mainly in the western part of Japan. The damage includes 216 fatalities, more than 20,000 damaged buildings and 8,500 houses inundated above floor level [6,7]. The most severe damage was recorded in Hiroshima prefecture including 108 fatalities,

six missing people, a total of 14,862 damaged buildings and 689 destroyed houses [8]. The damages in Hiroshima were caused mainly by debris flows occurred at approximately 8,000 sites.

## 2. Related Works and Research Objective

In order for emergency response, early-stage recovery and debris management to work efficiently following such landslide disasters, rapid identifications of affected areas and volumes of debris are indispensable. Remote sensing images such as aerial photographs and satellite images have contributed to detection of landslide areas over a wide area. Since trees and grass on the surface of slopes are collapsed and flowed to downstream in landslide events, vegetation indices such as the normalized difference vegetation index (NDVI) have been used as one of key parameters for landslide detection from remote sensing images [9–16]. Change detection using pre- and post-event images have been widely used for detecting areas affected by a disaster. In particular, pixel-based change detection techniques have been applied to middle- to high- resolution images for landslide detections [9–11]. More recently, object-oriented approaches have been introduced for landslide identification based on change detection using pre- and post-event images [17,18], change detections with topographical information derived from DEM [14], and fusion analysis of post-event image and DEM [13]. These object-oriented approaches would be effective, especially for very high-resolution images with the resolution of less than 1 m, because not only spectral information (tone, color), but also spatial features (size, shape, texture and pattern), can be incorporated in the identifications. However, in most of the object-oriented approaches, parameter tuning for image segmentation and selection of training samples are required to achieve accurate identification. On the contrary, although the pixel-based change detections are generally sensitive to the salt-and-pepper effect, fewer parameters are required than object-oriented approaches. Thus, they are still effective for the middle-resolution (around 10 m) images and for quick response after a disaster. In order to reduce false alarms, such as seasonal change of vegetation and mixed pixel-induced noises other than landslide-related changes, additional information would be necessary in the pixel-based approach.

Digital elevation models (DEMs) are also valuable data sources to evaluate topographical characteristics of the slopes. Recently, DEM-based debris flow susceptibility mapping methodologies have been developed for predicting and/or evaluating debris flow propagation areas [19–22]. Most of the debris flow simulation techniques require detailed soil parameters such as predefined volumes, soil thickness, weights, frictions and others. However, it is difficult to define such parameters for over a wide area before an event. Among the debris flow simulation techniques, Flow-R [20] can estimate regional-scale debris flow susceptibility mapping by providing predefined or automatically defined sources without assuming such detailed parameters of soil conditions. Therefore, the technique would be helpful to assist the rapid identification of debris flow propagation areas immediately after an event.

Light detection and ranging (LiDAR)-derived DEMs have been utilized for quantifying volumes of landslides [23–26]. The DEMs created from stereo pairs of satellite images have been also used for landslide volume estimations [27,28]. The volumes were estimated by aggregating the elevation change between pre- and post-event DEMs in landslide areas. However, in order to obtain such high-resolution DEMs, numbers of observations are required to cover whole affected areas in a large-scale disaster such as the July 2018 event in Hiroshima. If such LiDAR-derived DEMs are not available in affected areas, landslide areas need to be delineated manually in most cases from post-event aerial photographs before assessing the debris flow disaster. The preparation of the LiDAR-derived DEMs and the identification of landslide areas require a lot of time and labor. According to the previous studies for debris flow volumes [29–34], scaling relationships have been found between landslide areas and volumes. The results suggest the possibility of simplified volume estimation only from landslide areas by developing an empirical relationship between areas and volumes. In order for a rapid service for identifying landslide areas and quantifying volumes of debris flows, an automated or semi-automated technique without preparing post-event LiDAR data and manual delineation of landslide areas would be more effective.

Based on the backgrounds shown in Figure 1, this study proposes a fusion analysis technique of optical satellite images and pre-event DEM to detect landslide areas and quantify volumes of debris flows. First, an empirical erosion depth of debris flows is developed by analyzing elevation change between pre- and post-event LiDAR-derived DEMs observed in the affected areas by the debris flow disasters in Hiroshima. Second change detection technique is applied to identify landslide areas using vegetation indices between pre- and post-event satellite images. Third debris flow propagation analysis is performed by applying the regional-scale susceptibility mapping technique [Flow-R, 20] to pre-event DEM. Erosion areas due to the debris flows are extracted by combining the vegetation-based change detection and the debris flow propagation analysis. Volume is estimated by multiplying the developed empirical erosion depth to the detected erosion areas. Finally, the applicability of the technique is validated by comparing the estimated debris flow volumes with the LiDAR-derived volumes.

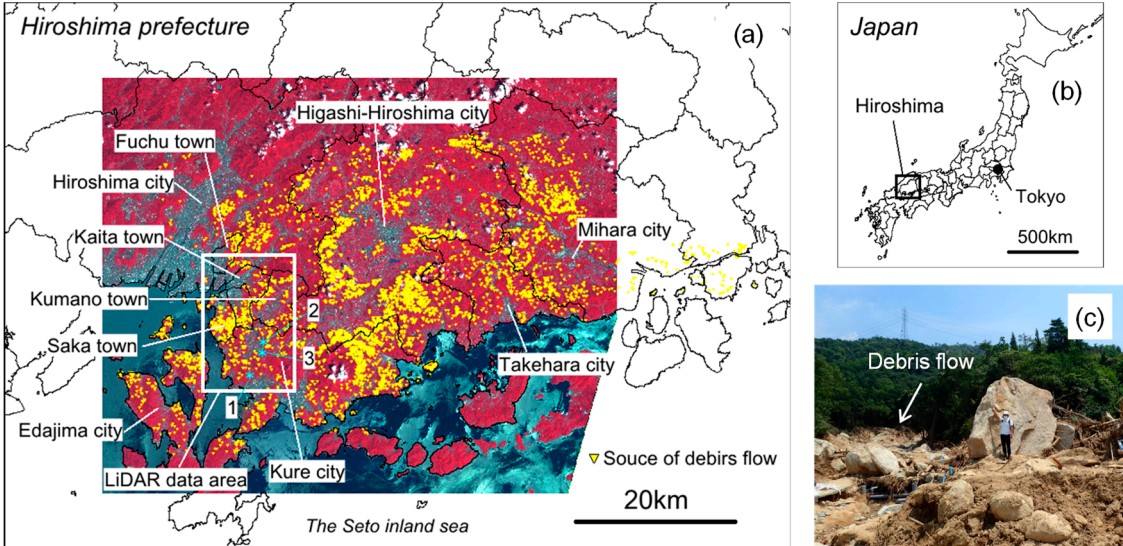

**Figure 1.** (**a**) Target area in Hiroshima prefecture, Japan. Background image is the post-event satellite Sentinel-2 image. Yellow triangles indicate sources of debris flows in the 2018 event [35]. White rectangular indicates LiDAR data area analyzed in this study. Blue stars (1-3) represent typical unaffected areas selected for accuracy assessment of LiDAR data. (**b**) Location of Hiroshima prefecture. (**c**) Ground photograph of debris flow with large blocks flowed during the 2018 event in Kumano town.

## 3. Methodology and Used Data

The target area of this study is the southern part of Hiroshima prefecture, Japan, severely damaged by the debris flows in the July 2018 event. Figure 1a,b show the location of Hiroshima. As shown by yellow triangles, debris flows were occurred at approximately 8,000 locations [35]. Most of the landslides collapsed from the evening of July 6 to the morning of July 7. During the event, maximum amounts of 72-hour rainfalls reached more than 400 mm in the affected areas [6]. Many buildings, roads and railways in Hiroshima city, Saka town, Kumano town, Kure city, Higashi-Hiroshima city, Takehara city and Mihara city were severely damaged by the debris flows. As shown in Figure 1c, large blocks with the size of larger than 3 m were flown to the residential area in Kuamano town.

Figure 2 illustrates the schematic diagram of a debris flow event. The collapse starts at a source point in a slope, and debris including soil, mud and rock flow to downstream along a valley thalweg. In the upstream area, the surface grounds are eroded due to the slope failures and the elevation decreases after the event. On the contrary, in the downstream area, the elevation increases after the event due to the debris deposition process. When the debris flow propagation area is divided to the erosion area and deposition area as shown in Figure 2, the erosion area can be detected from remotely sensed images because the collapse in the vegetated slope can be clearly identified from remote sensing images. In contrast, the identification of the deposition area is more difficult, because the discrimination

between the widely spread debris and other ground features such as buildings and bare grounds is not an easy task. Furthermore, the quantification of volumes from the deposition data is also difficult because it is almost impossible to quantify volumes flowed into affected buildings, even from LiDAR data, and extensive disposal of debris starts immediately after an event in affected residential area. On the other hand, since debris movement in an erosion area is more stable after a while from an event, reliable quantification of volumes can be achieved from the erosion data.

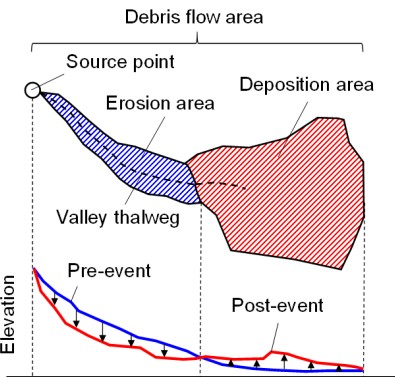

**Figure 2.** Schematic plan and cross-section of debris flow area.

In this study, a fusion analysis technique of satellite images and DEM is examined to identify erosion areas of debris flows. An empirical erosion depth model is developed from the relationship between erosion areas and volumes derived by LiDAR data analysis. A methodology for simplified quantifying debris flow volumes is proposed by combining the erosion area detection and the empirical erosion depth. Figure 3 shows the flowchart of the detection of erosion areas and the volume estimation for debris flows. First, elevation changes in erosion areas of debris flows are analyzed using the pre- and post-event LiDAR-derived DEMs in Hiroshima to obtain the empirical erosion depth model. Second, pixel-based change detection is applied using the pre- and post-event optical satellite images by NDVI differencing to extract the erosion candidate areas. Third, debris flow propagation analysis is performed by using the pre-event DEM and Flow-R [20] in order to extract the candidate areas affected by debris flows. Manually and automatically identified source points of the debris flows are examined in the analysis. Manual sources in the actual event are identified from the visual interpretation of the post-event aerial photographs. Automatic sources are extracted from the thresholds of slopes and upslope contributing areas proposed by Horton et al. [20]. Combining the result of the change detection, the debris flow propagation analysis and the slopes, erosion areas due to debris flows are detected. Finally, volumes of the debris flows are estimated from the detected erosion areas and the empirical erosion depth model.

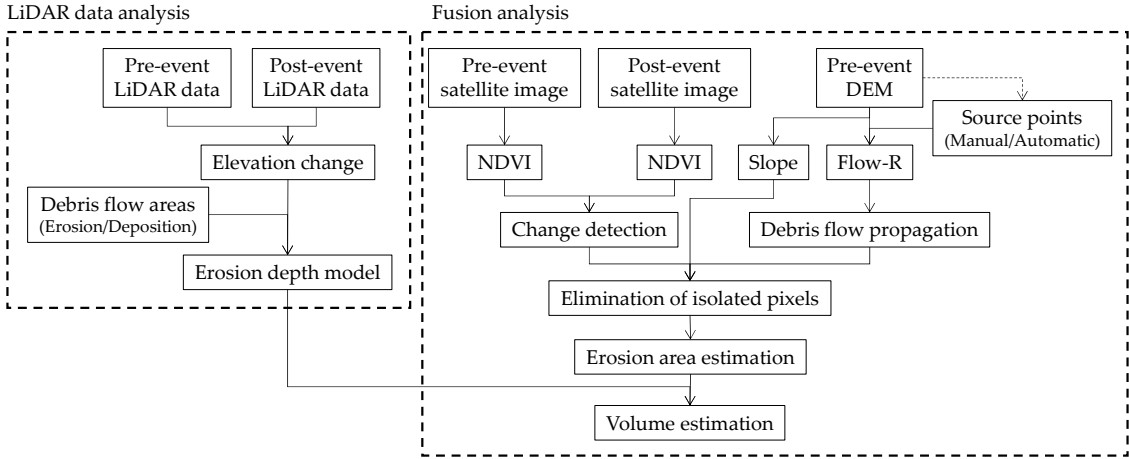

**Figure 3.** Flow of LiDAR data analysis and fusion analysis of optical satellite images and DEM.

The data list used in this study is summarized in Table 1. The data observed before and after the July 2018 debris flow disaster (this event hereafter is referred to as the 2018 event) in Hiroshima is mainly analyzed in this study. The high-resolution LiDAR-derived DEMs are provided by the Hiroshima prefectural office. The pre-event LiDAR was observed in 2015 to 2017. The post-event LiDAR was observed at 15 to 24 July 2018. The spatial resolution of the DEMs is 1 m. The coverage of the LiDAR data analyzed in this study is shown by a rectangle in Figure 1. The area covers the heavily damaged regions in the 2018 event such as the northern part of Kure city, Kumano town and Saka town. The optical satellite Sentinel-2 images [36] are used in this study because of their wide coverage, no cost and relatively high resolution. The pre- and post-event images were observed at 1 June and 16 July 2018, respectively. The visible and near infrared band images are analyzed. The spatial resolution of the images is 10 m. The DEMs developed by Geospatial Information Authority of Japan (GSI) [37] are used for the debris flow propagation analysis. The DEM in most of the target area was derived by LiDAR. Since LiDAR observation was not conducted in some area, the DEM was created by sequential aerial photo stereopairs. In this study, the DEMs derived from LiDAR and steraopairs were merged for the following analysis. These data was observed in 2015. Although the spatial resolution of the original DEM is 5 m, the resolution is converted to 10 m by down-sampling with cubic convolution technique to combine with the satellite image analysis and to reduce the computation time during the debris flow propagation analysis. The target area of this study is shown in Figure 1 as illustrated by the post-event Sentinel-2 image, covering most part of the affected areas by the 2018 event in Hiroshima prefecture.

**Table 1.** Characteristics of spatial data analyzed in this study. Spatial resolution of DEM for debris flow propagation analysis is converted to 10 m from 5 m by down-sampling.

| Data | Timing | Acquisition Date | Spatial Resolution (m) |
|---|---|---|---|
| LiDAR-derived DEM [*1] | Pre-event | 2015–2017 | 1 |
| | Post-event | 15–24 July 2018 | 1 |
| Optical satellite image (Sentinel-2 [*2]) | Pre-event | 1 June 2018 | 10 |
| | Post-event | 16 July 2018 | 10 |
| DEM derived by LiDAR and photogrammetry [*3] | Pre-event | 2015 | 10(5) |

Data source: [*1] Hiroshima prefecture; [*2] Copernicus (https://scihub.copernicus.eu/dhus/); [*3] Geospatial Information Authority of Japan (https://fgd.gsi.go.jp/download/).

## 4. LiDAR Data Analysis

As discussed in the previous studies [38,39], LiDAR-derived DEMs include uncertainties due to horizontal and vertical positional errors, LiDAR sampling density and filtering non-terrain features such as buildings and vegetation. Before analyzing the LiDAR-derived DEMs, vertical accuracy of the DEMs are assessed by comparing the elevations in areas unaffected by the debris flows. Three typical land surface such as slope, paddy field and built-up area are selected as shown in Figure 1 and the difference of elevations between the pre- and post-event DEMs are calculated. The subset area is approximately $500 \times 400$ m$^2$. Figure 4 shows the histograms of the difference of elevations in the three areas. The values in brackets indicate the mean value and standard deviation in each area, respectively. The standard deviation in the slope area is slightly larger than those in other areas. These standard deviations are almost comparable to the errors discussed in the previous studies [38,39].

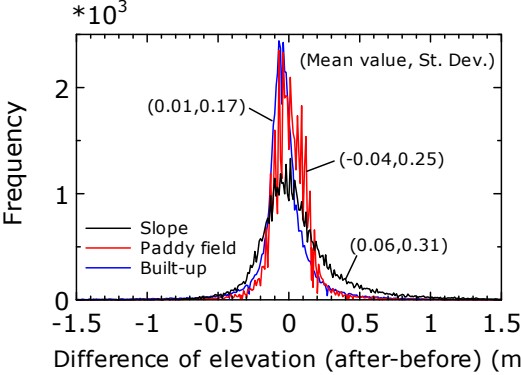

**Figure 4.** Histograms of difference of elevation between pre- and post-event LiDAR-derived DEMs in typical unaffected land surfaces.

The debris flow affected areas were delineated by visual interpretation of post-event aerial photographs and the polygon data is available in the website of the Association of Japanese Geographers [35]. Figure 5a shows the post-event aerial photograph in the affected area of Kumano town. The polygons indicate the interpreted debris flow affected areas. The attributes of erosion and deposition are assigned to each polygon by using the elevation change (after-before) obtained from the pre- and post-event LiDAR-derived DEMs as shown in Figure 5b. While the elevation is significantly decreased after the event in the erosion areas, the elevation is slightly increased in the deposition areas. Due to the errors in the LiDAR-derived DEMs shown above, we found elevation changes in the slopes other than the debris flow areas as shown in Figure 5b. This indicates that it would be still difficult to automatically identify the landslide areas without the help of aerial photographs.

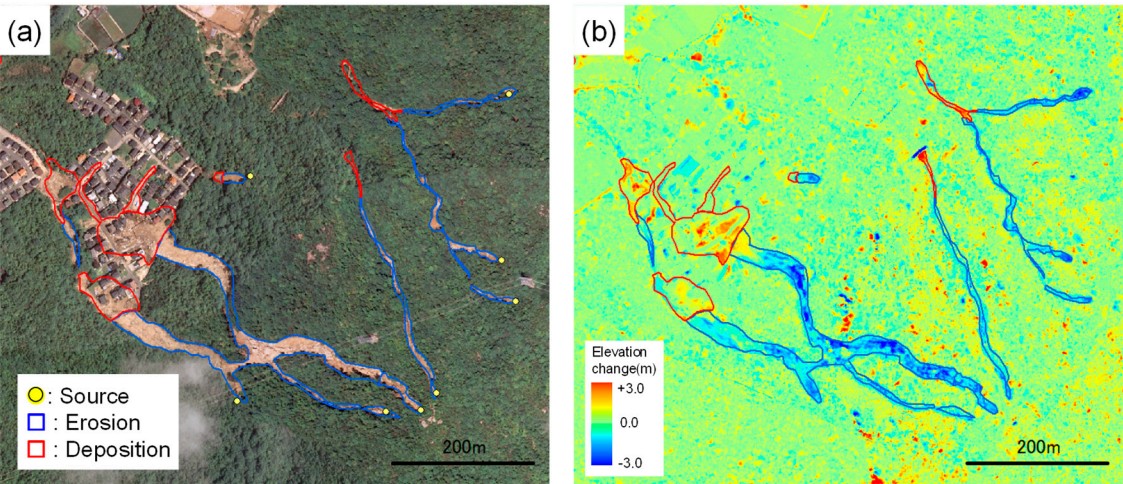

**Figure 5.** (**a**) Visually delineated erosion and deposition areas in debris flows in Kumano town. Background aerial photograph was observed after the event. (**b**) Elevation change (after-before) of LiDAR-derived DEMs.

The volumes of the debris flows are calculated by aggregating the negative elevation changes in each delineated erosion area. Figure 6a shows the relationship between area (*A*) and volume (*V*) derived from the erosion data of the 2018 event. The relationship obtained from the analysis of the LiDAR-derived DEMs in the debris flow event on 20 August 2014 in Hiroshima (hereafter the 2014 event) [5] is also plotted. Totally 637 debris flows are analyzed in this study. The data varies $1.6 \times 10^1$ m$^2 \leq A \leq 1.1 \times 10^5$ m$^2$ and $4.0 \times 10^1$ m$^3 \leq V \leq 8.6 \times 10^4$ m$^3$. The empirical relationships derived from the previous debris flow studies [29–33] summarized by Guzzetti et al. [34] are also illustrated in the figure. In the previous studies, the empirical relationships were modeled by an equation of the form $V = \alpha A^\gamma$. The equations, range of *A* and number of analyzed data of the previous studies are summarized in Table 2. Whereas the areas in the previous studies mean the total debris flow areas including erosion and deposition areas, this study limits only the erosion areas in the analysis. The data distributions of the 2014 and 2018 events are almost consistent with the previous empirical relationships. From a linear regression analysis based on least square approximation for the 2014 and 2018 events to derive the coefficients $\alpha$ and $\gamma$ of the form $V = \alpha A^\gamma$, the equation of *V-A* is developed as shown below.

$$V = 0.285A^{1.127} \tag{1}$$

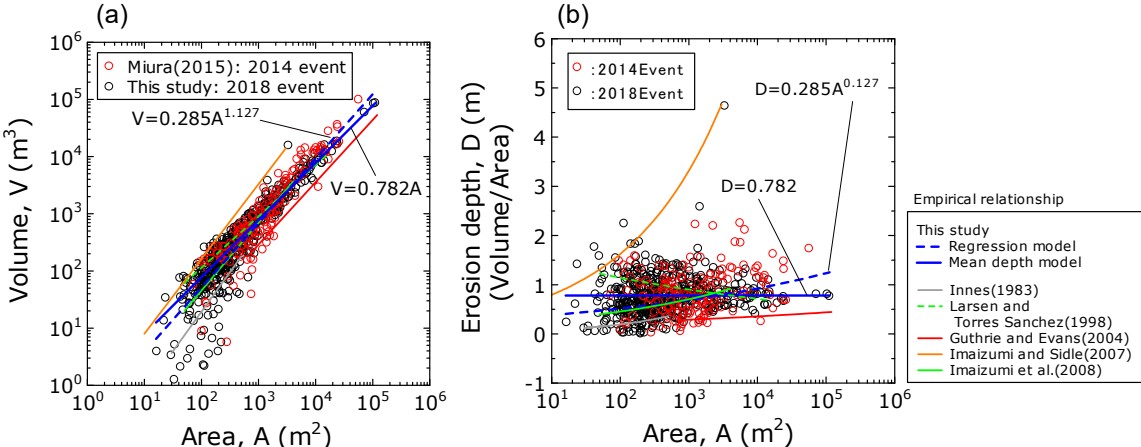

**Figure 6.** (**a**) Relationship between landslide area and volume of debris flow events. The relationships of the 2014 and 2018 events are derived from the LiDAR data analysis in Miura [5] and this study, respectively. (**b**) Relationship between erosion area and erosion depth derived from LiDAR data analysis in the 2014 and 2018 events. Empirical relationships derived in the previous debris flow studies are illustrated by colored lines. Empirical relationships derived from regression analysis and mean depth of the 2014 and 2018 events are plotted as blue dotted and solid lines, respectively.

**Table 2.** Empirical relationships linking area to volume obtained in this study and the previous studies [29–33]. Equation of Larsen and Torres-Sanchez [30] was developed by Guzzetti et al. [34].

| ID | Equation | Minimum A (m$^2$) | Maximum A (m$^2$) | N | Source |
|----|----------|-------------------|-------------------|---|--------|
| 1 | $V = 0.285A^{1.127}$ | $1.6 \times 10^1$ | $1.1 \times 10^5$ | 637 | Regression model (This study) |
| 2 | $V = 0.782A$ | | | | Mean depth model (This study) |
| 3 | $V = 0.0329A^{1.3852}$ | $3 \times 10^1$ | $5 \times 10^2$ | 30 | Innes (1983) |
| 4 | $V = 1.826A^{0.898}$ | $5 \times 10^1$ | $1.6 \times 10^4$ | 1019 | Larsen and Torres-Sanchez (1998) |
| 5 | $V = 0.1549A^{1.0905}$ | $7 \times 10^2$ | $1.2 \times 10^5$ | 124 | Guthrie and Evans (2004) |
| 6 | $V = 0.39A^{1.31}$ | $1 \times 10^1$ | $3 \times 10^3$ | 51 | Imaizumi and Sidle (2007) |
| 7 | $V = 0.19A^{1.19}$ | $5 \times 10^1$ | $4 \times 10^3$ | 11 | Imaizumi et al. (2008) |

The derived regression line is plotted in Figure 6a. The correlation coefficient between area and volume is 0.91. Since the dependency of volume on area is not significant in the equation ($\gamma = 1.127$), area-independent model ($\gamma = 1.0$) would be applicable for more simplified scaling.

In this study, average erosion depth is defined as erosion volume divided by erosion area. Figure 6b shows the relationship between the erosion areas and the depths obtained in the 2014 and 2018 events. The depth models from the regression analysis, including the previous empirical models shown above, are plotted as the exponential curves. Although the average erosion depths scatter is around 0 to 1.5 m, the variation of the data falls within the previous empirical relationships, and no significant dependency on the area is found for the erosion depths. The erosion depth can be approximated by an area-independent model. The mean value of the erosion depths is calculated at 0.78 m, as shown by the solid line. The area-independent erosion depth model (mean depth model) shown in Equation (2) would be more convenient for rapid and simplified volume estimation, because the total volume can be estimated directly from the pixel-based analysis without delineating landslide areas.

$$V = 0.782A \tag{2}$$

The empirical relationship of *V-A* by the mean depth model is also plotted in Figure 6a. The difference between the regression model and mean depth model is approximately 0.2 in logarithmic scale at maximum. Therefore, the mean erosion depth of 0.78 m is adopted for volume estimation in this study.

## 5. Fusion Analysis of Optical Satellite Images and DEM

### 5.1. NDVI-based Change Detection

The Sentinel-2 satellites were launched under the Copernicus Programme managed by the European Commission for the purpose of the optical imaging mission for land service [36]. The Sentinel-2A and Sentinel-2B launched on July 2015 and March 2017, respectively, are now operating. The pre- and post-event images were observed by the Sentinel-2A and the Sentinel-2B, respectively. The Level-1C products of the images contain orthorectification using a 90 m-grid DEM and an atmospheric correction of top-of-atmosphere [36]. In this study, the Level-2A images are developed from the Level-1C products by applying an atmospheric correction to obtain bottom-of-atmosphere reflectance products based on the Sen2Cor algorithm for the Sentinel-2 toolbox [40]. The Level-2A images are used in the following analysis. The Level-2A products also include automatically detected cloud and shadow mask. The pixels in the cloud and shadow masks of the pre- and post-event images are eliminated before analyzing.

Figure 7a,b show the close-up of the pre- and post-event Sentinel-2 images in the same area of Figure 5. False color images are displayed in the figures, indicating that the red colors represent strong reflections of vegetation in the near infrared band. The erosion areas generated by the debris flows can be easily identified from the decrease of the red color areas. In order to quantify the vegetation change, normalized difference vegetation index (NDVI) is calculated from each image based on the equation below.

$$NDVI = \frac{NIR - Red}{NIR + Red} \tag{3}$$

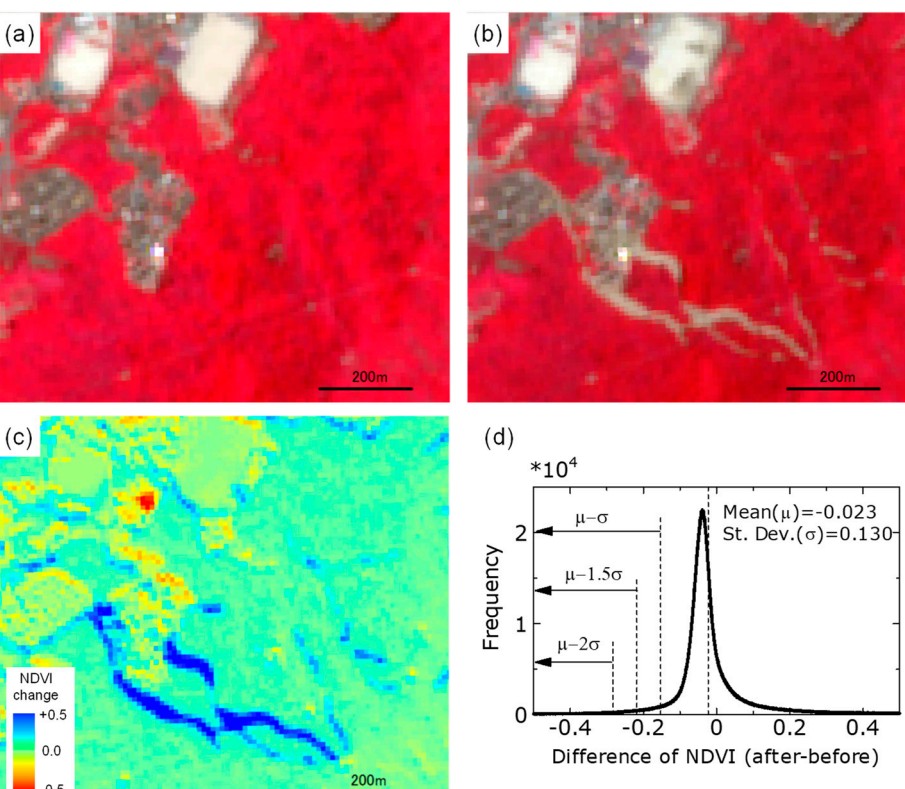

**Figure 7.** (**a**) Pre- event Sentinel-2 image in Kumano town. (**b**) Post-event Sentinel-2 image. (**c**) Distribution of NDVI change (after-before) with debris flow areas. (**d**) Histogram for difference of NDVI and thresholds for change detection.

Here, *NIR* and *Red* indicate the digital numbers of near infrared and red band images, respectively. NDVI is related to the amount of greenness in a pixel and yields a number from −1 to +1. A higher NDVI value means a higher density of vegetation. Pixel-based change detection is applied to the pre- and post-event NDVI images. Figure 7c shows the distribution of the NDVI change (after-before). We can clearly identify the significant decrease of NDVI in the erosion area.

A simple thresholding technique is applied to detect candidates of the erosion areas from the NDVI difference image. Figure 7d shows the histogram of the NDVI change. The data distribution can be approximated by normal distribution. The mean value ($\mu$) and the standard deviation ($\sigma$) are shown in the figure. The author examined the thresholds for pixel-based damage detection from optical images [41] and determined the threshold value at $\mu + 1.5\sigma$ by comparing detected pixels with ground truths. In this study, a preliminary analysis is performed by setting the three thresholds, namely $\mu - 1.0\sigma$ (T1), $\mu - 1.5\sigma$ (T2) and $\mu - 2.0\sigma$ (T3) as shown by arrows in Figure 7d. Since we found overestimation and underestimation detected by T1 and T3, respectively, T2 is selected as the threshold value for the erosion candidate detection. The pixels whose NDVI change is smaller than T2 are classified to erosion candidate areas.

## 5.2. Debris Flow Propagation Analysis

Only from the NDVI-based change detection, numbers of misdetections are extracted, such as seasonal change of the vegetation. In order to reduce such misdetections, numerous efforts have been undertaken by calibrating the geometric and radiometric characteristics and by including land use information [9–11,13,14,41]. In this study, the regional-scale debris flow propagation analysis, Flow-R [20] is performed to identify the candidates of debris flow affected areas including the erosion areas.

Flow-R is based on spreading the algorithm of flow accumulation. Flow-R can assess debris flow susceptibility by providing source areas, and subsequently simulating flow spreading based on multi-flow direction algorithm [42]. The simulation stops when the estimated debris flow velocity or travel angle in the distal zone reach a given criteria. Although several input parameters are still necessary for the propagation analysis, parameters for soil conditions that are difficult to define over a wide area are not required.

In the method, the debris flow susceptibility in each cell is calculated by the equations below:

$$p_i = \frac{p_i^{fd} p_i^{p}}{\sum_{j=1}^{8} p_j^{fd} p_j^{p}} p_0 \tag{4}$$

$$p_i^{fd} = \frac{(\tan \beta_i)^x}{\sum_{j=1}^{8} (\tan \beta_i)^x} \tag{5}$$

$$p_i^{p} = w_{\alpha(i)} \tag{6}$$

Here, *i* and *j* are flow directions. $p_i$ and $p_0$ indicate the susceptibility in a cell *i* and the susceptibility previously determined in a central cell, respectively. The susceptibility in a rupture starting cell is set at 1.0 in the calculation. $p_i^{fd}$ and $p_i^{p}$ represent the flow proportion according to the flow direction algorithm and flow proportion according to the persistence, respectively. $\tan \beta_i$ is the slope gradient between the central cell and the cell in direction *i*, *x* is the variable exponent. *w* and $\alpha(i)$ are the weight to the direction $\alpha(i)$ and the angle between the previous direction and the direction from the cell to cell *i* (0, 45, 90, 135 and 180).

The parameters for Equation (5) and (6) adopted in this study are summarized in Table 3. The modified version of the Holmgren method [42], introduced in Horton et al. [20], is used in the

flow direction algorithm. A factor *dh* (m), is given to the elevation of the central cell during calculating the gradients to avoid unexpected interruption by local convexity and concavity of the elevations. The simplified friction limited model characterized by a minimum travel angle is used to calculate the maximum possible runout distance. The minimum travel angle is set at 7 (degree). The maximum velocity is set at 15 (m/s), as not to exceed realistic velocities during the propagation analysis as shown in Table 3. The parameters for flow direction algorithm, travel angle and velocity are determined because they were commonly used for fine-grained flows in Horton et al. [20].

**Table 3.** Parameters and criteria used in Flow-R analysis.

| | | | | |
|---|---|---|---|---|
| Spreading Algorithm | Direction Algorithm | Holmgren (1994) Modified | $x$ | 4 |
| | | | $dh$ (m) | 2 |
| | Persistence | Weights, $w$ | | Default (Uniform weights 1.0 for all directions) |
| Energy calculation | Friction loss function | Threshold of travel angle(deg.) | | 7 |
| | Energy limitation | Maximum velocity(m/s) | | 15 |

The applicability of the parameters were already examined to reproduce the maximum runout distance and propagation areas of the debris flows in the 2014 event in Hiroshima [43]. The typical debris flow area and the damaged area in the 2014 event is shown in Figure 8a. Figure 8b shows the result of the debris flow propagation based on the parameters in Table 3. The extent and maximum runout distance of the event were slightly overestimated, but almost successfully reproduced by the criteria. On the other hand, Figure 8c shows the result of the propagation when the flow proportion weights for persistence, $p_i^p$ are changed to the Gamma's criteria [44] introduced in Horton et al. [20]. As shown in Table 4, the Gamma's criteria provide larger weight for straightforward propagation ($w_0 = 1.5$) and do not allow backward propagation ($w_{180} = 0$). The results show that the Gamma's criteria significantly underestimated the actual debris flow area because the wide spreading in the 2014 event was not reproduced by the criteria.

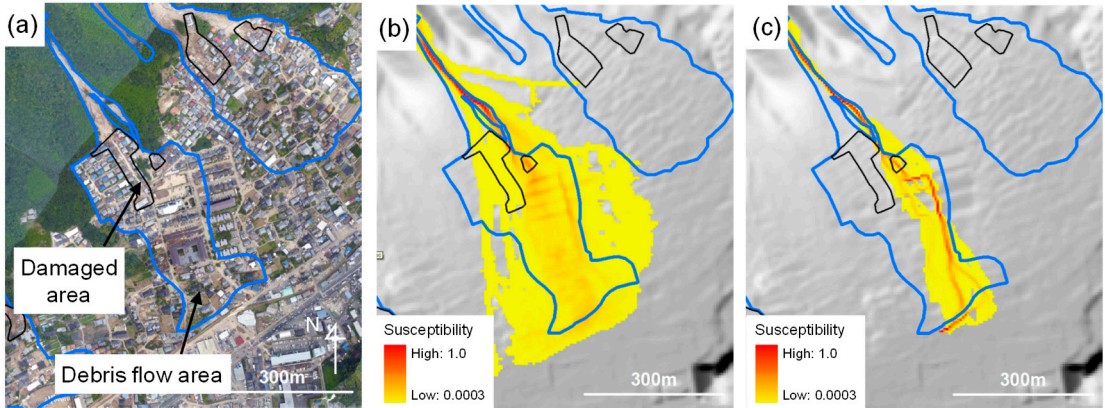

**Figure 8.** (**a**) Debris flow area with post-event aerial photograph observed after the 2014 event in Hiroshima. (**b**) Debris flow propagation area by Flow-R analysis based on the parameters in Table 3. (**c**) Debris flow propagation area by Flow-R analysis based on Gamma's criteria [44] for weights in persistence function as shown in Table 4.

**Table 4.** Comparison of weights for persistence function in spreading algorithm in Equation (6).

| | $w_0$ | $w_{45}$ | $w_{90}$ | $w_{135}$ | $w_{180}$ |
|---|---|---|---|---|---|
| Default | 1 | 1 | 1 | 1 | 1 |
| Gamma [2000] | 1.5 | 1 | 1 | 1 | 0 |

Two kinds of source areas are provided in the debris flow propagation analysis. One is the manual source points visually interpreted from the aerial photographs in the actual 2018 events. The other is automatic source areas automatically detected from the criterion proposed in Horton et al. [20]. Two criterion were proposed for extreme and rare events as shown in Equations (7) and (8), respectively.

$$\begin{cases} \tan\beta_{thres} = 0.32S_{uca}^{-0.2} & \text{if} \quad S_{uca} < 2.5 \text{ km}^2 \\ \tan\beta_{thres} = 0.26 & \text{if} \quad S_{uca} \geq 2.5 \text{ km}^2 \end{cases} \quad (7)$$

$$\begin{cases} \tan\beta_{thres} = 0.31S_{uca}^{-0.15} & \text{if} \quad S_{uca} < 2.5 \text{km}^2 \\ \tan\beta_{thres} = 0.26 & \text{if} \quad S_{uca} \geq 2.5 \text{ km}^2 \end{cases} \quad (8)$$

Here, $\tan\beta_{thres}$ is the slope threshold, and $S_{uca}$ is the upslope contribution area. Although the manual source points are expected to reproduce more accurate debris flow propagation areas, the visual identification of the sources would be time-consuming. If source areas can be automatically and accurately determined without visual interpretation of aerial photos, rapid debris flow assessment can be achieved. In order to examine the applicability of the automatically defined sources for assessing the actual debris flow areas, the debris flow propagation areas are estimated not only by using the manually defined sources, but also by using two automatically defined source areas. The automatically defined source areas are determined only by the topographical characteristics in this study, and do not consider other external factors such as site-dependent rainfall intensities. This implies that all the potential sources would be simultaneously collapsed in whole the target area.

Figure 9a shows the estimated propagation areas by the manually identified source points. Compared to the debris flow areas of the 2018 event, the erosion areas are correctly simulated by the analysis. On the contrary, the deposition areas are mostly overestimated, because the analysis does not consider the volume in each collapse and the parameters are set to reproduce the maximum runout. Figure 9b shows the estimated propagation areas by the automatic source areas based on the criteria for extreme event in Equation (7). Since the source areas determined from this criteria are located at much lower part of the slopes than the 2018 event, the erosion areas are largely underestimated. This means that the much more severe criteria is required to reproduce the sources of the 2018 event. When the criteria for rare events shown in Equation (8) are given for source estimation, source areas are not detected in this area, indicating that this criteria is far from sufficient to reproduce the debris flows in this area.

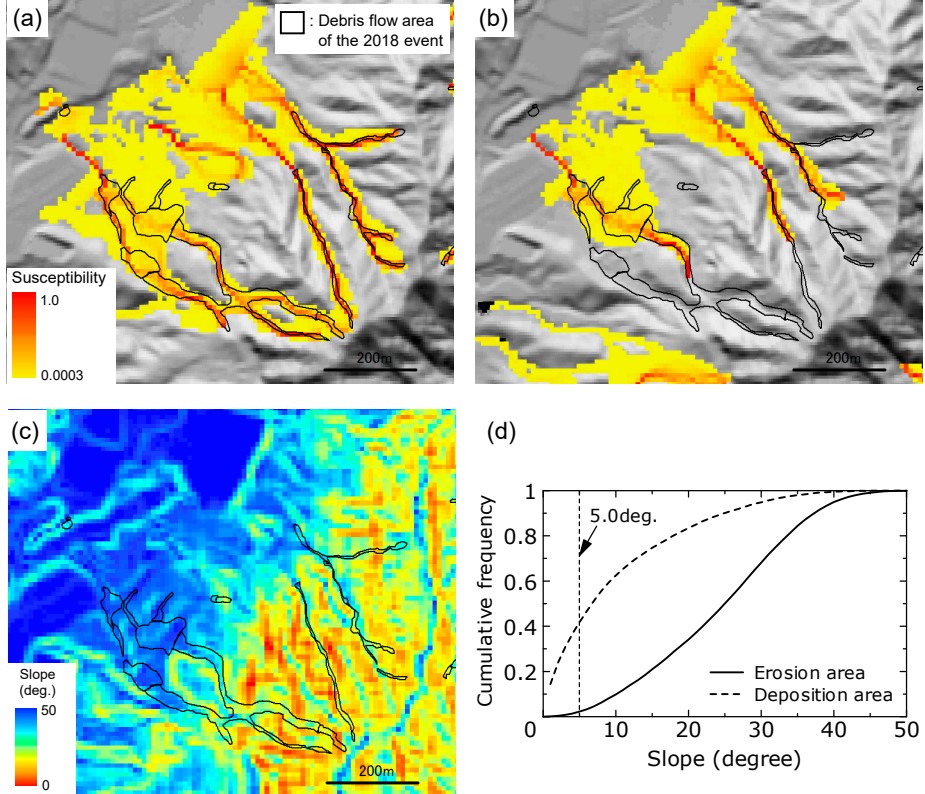

**Figure 9.** (**a**) Debris flow propagation areas by Flow-R analysis using manually identified sources. (**b**) Debris flow propagation areas using automatically identified sources based on criteria for extreme event. (**c**) Slope gradients with debris flow areas. (**d**) Cumulative frequency of slopes in erosion and deposition areas.

Figure 9c shows the distribution of the slope gradients of the DEM. The number of pixels are aggregating to count the slopes in the erosion and deposition areas. Figure 9d shows the cumulative frequencies aggregated in the erosion and deposition areas. Since slope for approximately 98% in the erosion area is higher than 5 (deg.), the threshold value for slope is set at 5 (deg.) in order to eliminate misdetection in flat zones.

## 6. Estimation of Debris Flow Volumes

Combining the NDVI-based change detection and the Flow-R-based propagation analysis, the erosion areas created by debris flows are detected. When the pixels are extracted by the change detection, estimated by the debris flow propagation analysis and steeper than 5 degree, they are classified to the erosion areas. Finally, isolated pixels defined as a detected pixel surrounding all by undetected pixels are eliminated from the results. Figure 10 shows the distribution of the detected pixels using the manual source points in the Flow-R analysis. Approximately 5% of the detected pixels are reduced by the elimination in this case. Although some pixels are over-detected in the downstream, most part of the erosion areas are successfully detected by the fusion analysis. The results based on the manual source points are mainly discussed in the following analysis. The volumes are estimated by multiplying the mean value of the erosion depth (0.78 m) to the detected areas. Figure 11 shows the distribution of the estimated volumes aggregated by 250 m mesh map. Large number of volumes are estimated in the southern part of Higashi-Hiroshima city, the western part of Kure city, Saka town and Kumano town. The spatial pattern of volumes generated by the debris flows is clearly identified, and it would be useful for early-stage debris management planning.

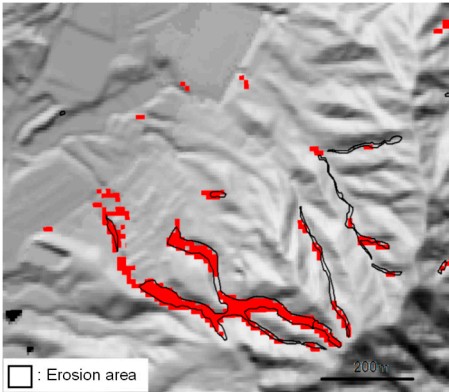

**Figure 10.** Detected pixels (red) by the fusion analysis in this study with erosion areas visually identified from aerial photographs.

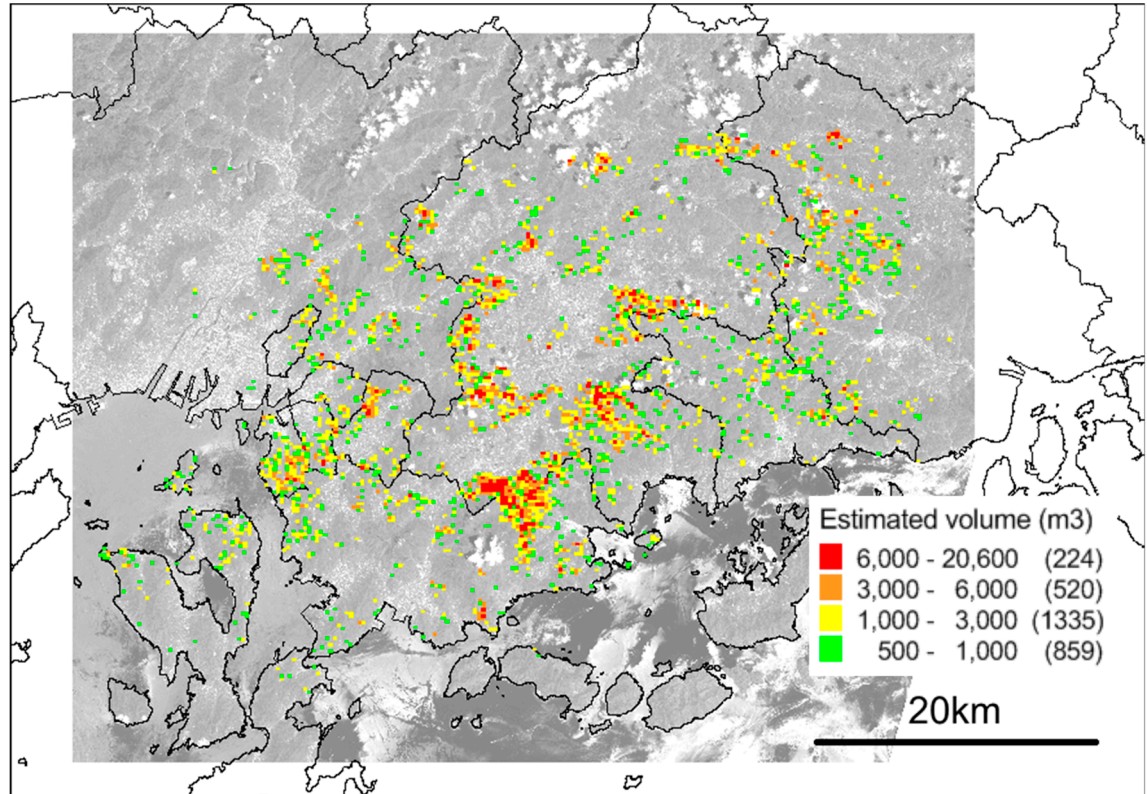

**Figure 11.** Distribution of volumes estimated in this study. Manually identified sources in Flow-R analysis and mean erosion depth model are adopted in this analysis. Volumes are summarized by 250 m-mesh map.

The Hiroshima prefectural office performed the LiDAR observations over the target area after the 2018 event [45]. The volumes mobilized by the landslides were evaluated by aggregating the elevation change between the pre- and post-event LiDAR-derived DEMs in the heavily damaged areas. In some areas where LiDAR data was not available, they estimated the volumes with the following procedure: first, they delineated the extents of the debris flow areas and their thalweg lines from aerial photographs for the whole target area. Second, they examined the relationship between the length of the thalweg line and volume for each debris flow, and derived mean volume value per unit length of thalweg line from the LiDAR data. Finally, they estimated the volumes where the LiDAR data was not available from the identified thalweg line length by multiplying the mean volume per unit length. The volumes obtained from these observations were summarized for each municipality.

Since most of the severe debris flows were captured by the LiDAR observations, the result of the Hiroshima prefectural office is hereafter referred to as LiDAR-derived volumes. Figure 12 shows the comparison of the volumes in each municipality between the LiDAR observations and the estimation of this study based on the manual sources in the Flow-R analysis. As shown in Table 5, the total volume derived from the LiDAR observations was 7.50 million cubic meters. The estimate of the total volume in this study is 7.35 million cubic meters. Although the estimation of this study is in total slightly underestimated, the estimated volumes show good agreement with the observed volumes, not only in total, but also in each municipality as shown in Figure 12 and Table 5, indicating the validity of the proposed method.

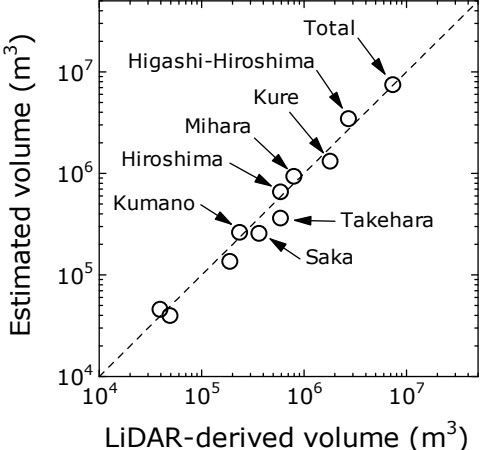

**Figure 12.** Comparison of LiDAR-derived volumes and estimated volumes based on manual sources in Flow-R analysis for each municipality and total.

**Table 5.** Comparison of volumes in each municipality between LiDAR-observation and estimations of this study.

| ID | Municipality | | Volume (Million Cubic Meters, ×10⁶ m³) | | | | |
|---|---|---|---|---|---|---|---|
| | | | Observed | Estimated | | | |
| | | | LiDAR * | Manual Source | Without Flow-R | Automatic Source | |
| | Name | Area (sq km) | | with Flow-R | | Extreme Event | Rare Event |
| 1 | Hiroshima city | 902 | 0.60 | 0.65 | 7.32 | 1.19 | 0.76 |
| 2 | Kure city | 352 | 1.83 | 1.30 | 3.48 | 0.82 | 0.44 |
| 3 | Higashi-Hiroshima city | 636 | 2.77 | 3.40 | 10.89 | 1.47 | 0.57 |
| 4 | Takehara city | 118 | 0.60 | 0.36 | 1.87 | 0.22 | 0.12 |
| 5 | Mihara city | 471 | 0.81 | 0.92 | 4.08 | 0.30 | 0.15 |
| 6 | Edajima city | 100 | 0.19 | 0.13 | 0.41 | 0.08 | 0.05 |
| 7 | Kumano town | 34 | 0.24 | 0.26 | 0.35 | 0.10 | 0.04 |
| 8 | Saka town | 16 | 0.37 | 0.25 | 0.34 | 0.16 | 0.11 |
| 9 | Fuchu town | 10 | 0.05 | 0.04 | 0.10 | 0.02 | 0.01 |
| 10 | Kaita town | 14 | 0.04 | 0.05 | 0.09 | 0.04 | 0.03 |
| | Total | 2653 | 7.50 | 7.35 | 28.93 | 4.40 | 2.28 |

*: Observed by Hiroshima prefecture [45].

In order to examine the effect of the debris flow propagation analysis by the Flow-R, the volumes are estimated without using the result of the Flow-R analysis. Figure 13a shows the comparison of the results with and without the Flow-R analysis. The result without Flow-R analysis largely overestimate the volumes due to the over-extraction by the NDVI-based change detection. This indicates that the Flow-R analysis contributes to the accurate identification of the erosion areas. Next in order to examine the effect of the defined sources in the Flow-R analysis, the volumes are also estimated from the fusion analysis based on the automatic source areas in Flow-R. Figure 13b shows the estimation

results by the automatic sources based on the criteria for extreme and rare events. The results by the automatic sources are significantly smaller than the observation especially by the criteria for rare event. The results of these estimations are summarized in Table 5. This indicates that the criteria in Equations (7) and (8) are insufficient for reproducing the debris flows of the 2018 event. These results show that the identification of the sources in debris flow propagation analysis would be key for accurate volume estimation. Although the identification of the sources is significantly less time-consuming than the delineation of the landslide areas, the manual identifications over a wide area such as the 2018 event still requires at least several days [39]. Accurate and automatic estimation of the debris flow sources, based not only on topographical characteristics but also on site-dependent rainfall intensity, needs to be developed for more rapid volume estimation.

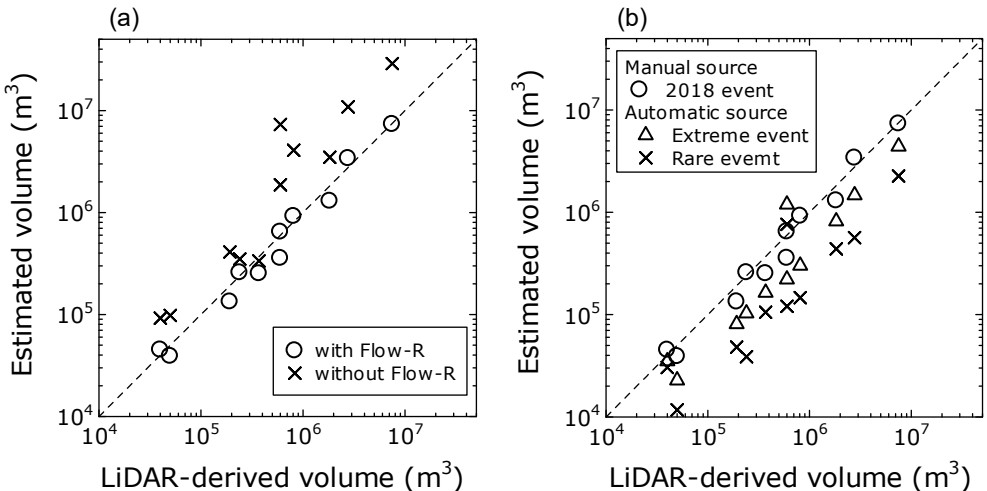

**Figure 13.** (**a**) Comparison of volume estimations with and without using Flow-R-based propagation analysis. (**b**) Comparison of volume estimations based on manual sources and automatic sources in Flow-R analysis.

## 7. Conclusions

This study proposes a methodology for fusion analysis of optical satellite images and DEM to detect the erosion areas by debris flow and to rapidly estimate the volumes. From the analysis of the pre- and post-event LiDAR-derived DEMs in the affected area by the debris flow events in Hiroshima, the empirical relationships between the erosion areas and the debris volumes were developed. The relationship was almost consistent with the previous debris flow studies. The area-independent mean erosion depth model was derived for estimating volumes from erosion areas. Pixel-based change detection approach was applied to the optical satellite Sentinel-2 images observed before and after the event. From the NDVI-based change detection, the vegetation decreased areas were detected as candidates of the erosion areas by debris flows. The debris flow propagation analysis was also performed using the pre-event DEM. The erosion areas in the 2018 event were correctly reproduced by the Flow-R-based propagation analysis using the manually identified source points. Combining the results, the erosion areas were detected and the volumes were estimated by applying the mean erosion depth to the detected pixels. The estimated volumes showed good agreement with the volumes derived from the LiDAR data.

The proposed methodology provides rapid volume estimation for debris flow disaster without LiDAR observation and manual delineation of debris flow areas. However, visual interpretation of debris flow sources is still required for accurate estimation. In order for a more rapid estimation, a methodology for accurate and automatic identification of debris flow sources needs to be developed in future work.

**Author Contributions:** H.M. conceived the research framework, analyzed the data and wrote the manuscript.

**Funding:** This study is partially supported by JSPS Kakenhi (Grant No. 17H02050 and 19H02408).

**Acknowledgments:** The LiDAR-derived DEMs were provided by the Hiroshima prefectural office. The Sentinel-2 images and the pre-event DEM were downloaded from the websites of Copernicus Open Access Hub (https://scihub.copernicus.eu/dhus/#/home) and Geospatial Information Authority of Japan (https://fgd.gsi.go.jp/download/menu.php), respectively. The debris flow area polygon data was downloaded from the website of The Association of Japanese Geographers (http://ajg-disaster.blogspot.com/2018/07/3077.html).

**Conflicts of Interest:** The author declares no conflict of interest.

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
