# Peer review of "Fusion Analysis of Optical Satellite Images and Digital Elevation Model for Quantifying Volume in Debris Flow Disaster"

_remotesensing, doi:10.3390/rs11091096_

Round 1
Reviewer 1 Report
This manuscript presents a methodology for quantifying the volume of debris flows by fusing optical satellite imagery and digital elevation models.
The method is partially innovative and introduces a straightforward application that integrates the results of two different analysis, NDVI change detection and debris flow simulation using Flow-R packages. The experiment is well designed and allows the readers to see the advantages of fusing earth observation data and computer simulation.
There are, however, some issues that need further clarification before publication. This manuscript is well written; however, it requires some revision of the following comments.
1) The literature review is somewhat limited. The author mainly focuses on change detection using spectral bands index. There are, however, other techniques are using remote sensing data for estimating landslide areas worth mentioning. For instance:
- Semiautomatic object-oriented landslide recognition scheme from multisensor optical imagery and DEM
- Landslide Inventory Mapping From Bitemporal High-Resolution Remote Sensing Images Using Change Detection and Multiscale Segmentation
2) LiDAR Data Analisis (L165-168), It would be interesting to see a comparison of the pre- and post-event LiDAR data (perhaps in Fig. 4 which is missing). A quantitative comparison may improve the discussion since DEM derived from LiDAR measurements should be accurate enough to detect significant land surface changes.
3) NDVI-based change detection: This section has a critical issue. The author emphasises that Sentinel-2 Level-1C products, without further preprocessing (L205-207), wee used for index analysis. Cording to the specs (https://earth.esa.int/web/sentinel/user-guides/sentinel-2-msi/product-types/level-2a), only top-of-atmosphere reflectance is provided in Level-1C images; these images are not suitable for change detection analysis since atmospheric-correction is still necessary to avoid illumination anomalies. At least Level-2A which has bottom-of-atmosphere reflectance is required.
4) Debris flows propagation analysis: The author mention (L245-247) that Flow-R is applicable for simulating debris flow in Hiroshima, Japan (reference No.34), based on a previous event in 2014. It seems that the paper reference describes the parameters tunning for Flow-R model. However, the cited paper only has an English abstract; thus it may be difficult for a non-Japanese speaker to follow this research. The main idea of citing previous works is to give readers the ability to reproduce this work. Since the author published the reference No.34, it would be helpful if a summary of the main Flow-R settings for Hiroshima is included in the manuscript.
5) Estimation of debris flow volumes: I would suggest revising if the sentences in lines 333-335 still hold after verifying the Sentinel-2 product levels.
6) Table 1 is also missing in the manuscript.
Author Response
Thank you for gentle comments for improving my study. The point-by-point responses for your comments are shown in the attached file.

Reviewer 2 Report
Dear Author,
the paper you presented is interesting but there are major issues that should be addressed befor publication. First of all: it is not possible to receive a manuscript where tables and figures are missing
Then other broader comments:
· References are outdated, new literature about similar approaches is available and should be cited
· How does the 10 m resolution for the simulation do you think affects your analysis?
· What are the correlation coefficients of your regression?
· What is the take home message from your work?
· Some sentences are not clear, the English of the paper should be checked again
Lots of more comments are in the annotated attached pdf.

Author Response

(The authors gave the same response as above.)

Round 2
Reviewer 2 Report
Dear Author
thank you for incorporating most of my advices.
I think that your paper now may be published
Best regards
Giulia Bossi